# The Role of *Mycoplasma bovirhinis* in the Development of Singular and Concomitant Respiratory Infections in Dairy Calves from Southern Brazil

**DOI:** 10.3390/pathogens13020114

**Published:** 2024-01-26

**Authors:** Ana Paula Souza Frucchi, Alais Maria Dall Agnol, Eloiza Teles Caldart, Dalton Everton Bronkhorst, Alice Fernandes Alfieri, Amauri Alcindo Alfieri, Selwyn Arlington Headley

**Affiliations:** 1Laboratory of Animal Virology, Department of Preventive Veterinary Medicine, Universidade Estadual de Londrina, Londrina 86057-970, Brazil; ana.paula.frucchi@uel.br (A.P.S.F.); alaisagnol@uel.br (A.M.D.A.); daltonbronkhorst@hotmail.com (D.E.B.); aalfieri@uel.br (A.F.A.); alfieri@uel.br (A.A.A.); 2Laboratory of Protozoology and Parasitic Diseases, Department of Preventive Veterinary Medicine, Universidade Estadual de Londrina, Londrina 86057-970, Brazil; eloizacaldart@uel.br; 3Multi-User Animal Health Laboratory (LAMSA), Department of Preventive Veterinary Medicine, Universidade Estadual de Londrina, Londrina 86057-970, Brazil; 4National Institute of Science and Technology for Dairy Production Chain (INCT–LEITE), Universidade Estadual de Londrina, Londrina 86057-970, Brazil; 5Laboratory of Animal Pathology, Department of Preventive Veterinary Medicine, Universidade Estadual de Londrina, Londrina 86057-970, Brazil

**Keywords:** bovine coronavirus, bovine respiratory disease, infection dynamics, *Mycoplasma* spp., ovine gammaherpesvirus 2

## Abstract

The role of *Mycoplasma bovirhinis* in the development of pulmonary disease in cattle is controversial and was never evaluated in cattle from Latin America. This study investigated the respiratory infection dynamics associated with *M. bovirhinis* in suckling calves from 15 dairy cattle herds in Southern Brazil. Nasal swabs were obtained from asymptomatic (*n* = 102) and calves with clinical manifestations (*n* = 103) of bovine respiratory disease (BRD) and used in molecular assays to identify the specific genes of viral and bacterial disease pathogens of BRD. Only *M. bovirhinis*, bovine coronavirus (BCoV), ovine gammaherpesvirus 2 (OvGHV2), *Histophilus somni*, *Pasteurella multocida*, and *Mannheimia haemolytica* were detected. *M. bovirhinis* was the most frequently diagnosed pathogen in diseased (57.8%; 59/102) and asymptomatic (55.3%; 57/103) calves at all farms. BCoV-related infections were diagnosed in diseased (52%; 53/102) and asymptomatic (51.4%; 53/103) calves and occurred in 93.3% (14/15) of all farms. Similarly, infectious due to OvGHV2 occurred in diseased (37.2%; 38/102) and asymptomatic (27.2%; /28/103) calves and were diagnosed in 80% (12/15) of all farms investigated. Significant statistical differences were not identified when the two groups of calves were compared at most farms, except for infections due to OvGHV2 that affected five calves at one farm. These results demonstrated that the respiratory infection dynamics of *M. bovirhinis* identified in Southern Brazil are similar to those observed worldwide, suggesting that there is not enough sufficient collected data to consider *M. bovirhinis* as a pathogen of respiratory infections in cattle. Additionally, the possible roles of BCoV and OvGHV2 in the development of BRD are discussed.

## 1. Introduction

Bovine respiratory disease (BRD) is a multifactorial and multietiological disease complex associated with a wide range of infectious disease pathogens and abrupt alterations to management practices and environmental conditions [1,2,3]. Infectious agents associated with BRD include conventional respiratory pathogens of ruminants such as *Histophilus somni*, *Pasteurella multocida*, *Mannheimia haemolytica*, *Mycoplasma bovis*, the bovine viral diarrhea virus (BVDV), bovine alphaherpesvirus 1 (BoAHV1), bovine respiratory syncytial virus (BRSV), bovine coronavirus (BCoV), and bovine parainfluenza virus 3 (BPIV3) [3,4,5,6]. Additional infectious agents considered to be associated with the development of BRD include the ovine gammaherpesvirus 2 (OvGHV2) [7], bovine adenovirus 3, influenza D virus, and bovine rhinitis A virus [8,9].

The role of *Mycoplasma bovirhinis* (*M. bovirhinis*) in the development of BRD is controversial since this organism was identified in the lungs of cattle without any association with pulmonary disease [8,10]. Nevertheless, a significant predominance of *M. bovirhinis* was identified in the aspirated bronchoalveolar fluid [11] and the nasopharyngeal microbiota [12] of diseased calves relative to calves without clinical manifestations of BRD, suggesting a possible role of *M. bovirhinis* in the development of BRD. Additionally, *M. bovirhinis* was indicated as the etiologic agent of an outbreak of pneumonia in calves [13], seems to be endemic in ruminants from the UK [14,15], and was identified in pneumonic calves from Holland [16]. Furthermore, *M. bovirhinis* was the predominant pathogen detected in the nasal swabs (NS) of calves and was associated with an outbreak of BRD in the USA [17]. However, *M. bovirhinis* was one of the most frequently identified organisms from the nasopharyngeal microbiota of calves [18,19,20] and was identified in almost equal proportions in asymptomatic, diseased, and calves suspected of BRD [10,21]. These findings suggest that *M. bovirhinis* may or may not be associated with the development of BRD in cattle. Consequently, additional studies are required to investigate the possible participation, if any, of *M. bovirhinis* in the development of pulmonary disease in cattle.

It must be highlighted that all previous studies that have investigated the possible association of *M. bovirhinis* in the development of BRD were performed either in North America [12,17,18,19,20], Europe [16,21], the UK [14], or Asia [13]. Similar investigations were not identified in Latin or South America when major databases were searched. Consequently, there is a need to understand the dynamics of this agent with the association of BRD in cattle from South America, considering that the existing environmental differences [22], as well as the differences in the farming and management systems [23], used in these distinct geographical regions may have some influence on the occurrence, development, and maintenance of the disease in animal populations. Therefore, we investigated the possible role of *M. bovirhinis* in the development of BRD in dairy calves from Southern Brazil.

## 2. Materials and Methods

### 2.1. Animals and Study Location

The occurrence of respiratory disease was investigated in 15 dairy farming units located in the rural districts of several cities, predominantly within the East-West mesoregion of the state of Paraná, Southern Brazil. Part of the samples were from a previous study [24]. These farms consisted of small to medium-sized dairy establishments that are maintained essentially for milk production. Suckling calves at these farms, consisting predominantly of the Holstein breed, were not immunized against any infectious disease agent and were maintained in collective pens. All calves received commercially produced milker replacements and/or milk produced on each farm and water *ad libitum*. 

### 2.2. Sampling and Inclusion Criteria 

The sampling was performed during on-site visits to these establishments and before the initiation of any therapeutic intervention. Nasal swabs (NS) were obtained from all animals, maintained on ice, and submitted for laboratory evaluation within four hours after collection. All NS were collected with commercially produced synthetic nylon swabs (16 cm) that were inserted as deep as possible into the ventral meatus of the nostril. The nostrils of all calves were cleaned with disposable paper towels before sampling.

Calves were characterized as diseased due to the demonstration of a combination of at least four of the following clinical manifestations indicative of BRD: fever (>40 °C), cough, nasal and/or ocular discharge, and dyspnea [25]. Dairy calves without any of these clinical manifestations were classified as asymptomatic. Furthermore, calves with at least one of the clinical manifestations were not included in this study. Accordingly, NS were obtained from diseased (*n* = 103) and asymptomatic (*n* = 102) dairy calves and then used in molecular assays designed to detect the possible association of conventional and nonconventional agents associated with the development of BRD. 

### 2.3. Molecular Detection of Agents Associated with the Development of BRD

Nucleic acids were extracted from all NS using a combination of the phenol/chloroform/isoamyl alcohol and silica/guanidine isothiocyanate methods as described [26,27]. The extracted nucleic acids were then used in molecular assays that targeted specific genes of *H. somni* [28], *P. multocida* [29], *M. haemolytica* [11], *M. bovis* and mollicutes [30], BVDV [31], BoAHV1 [32], BRSV [33], BCoV [34], BPIV3 [35], and OvGHV2 [36]. A list of the specific target genes of these organisms with the primer sequences and the desired amplicon size in base pairs (bp) is provided (Appendix A Appendix A). Positive controls consisted of the utilization of DNA/RNA of these organisms derived from previous studies [7,37]. Sterile, ultrapure water (Invitrogen Life Technologies, Carlsbad, CA, USA) was used as a negative control. Positive and negative controls were included in all molecular assays.

### 2.4. Molecular Identification of Mycoplasma Bovirhinis

The detection of *M. bovirhinis* was performed due to the development of specific primers that were used in a two-step process. Initially, the primers (MolliF/MolliR) were used for the amplification of the 16S-23S rRNA internal transcribed spacer (ITS) region of the mollicute genome [30]. This consensual PCR assay was designed to detect mollicutes and amplify PCR products with lengths ranging from 847 to 866 bp. All PCR conditions and procedures were performed as previously described [30]. 

The detection of *M. bovirhinis* was performed via the utilization of primers (forward 5′-ATAAGGTTATCATTTCTTATT-3′ and reverse 5′-AAAATAGTCTTGAATGCG-3′) designed for the specific amplification of 419 bp internal to the previously mentioned PCR assay [30]. These primers were then used in a second round of PCR (nested-PCR, nPCR). All reactions were performed with final solutions of 50 µL containing a 1× PCR buffer (20 mM Tris-HCl pH 8.4 and 50 mM KCl), 2.5 mM MgCl_2_, 10 mM of each dNTP, 20 pmol of each primer (forward and reverse), 2.5 U Platinum Taq DNA Polymerase (Invitrogen™ Life Technologies, São Paulo, SP, Brazil), and 1 µL PCR product. Ultrapure sterile water was added to a final volume of 50 μL. Amplification was performed with the following cycling profile: an initial step of 5 min at 94 °C, followed by 35 cycles of 30 s/94 °C, 30 s/48 °C, 1 min/72 °C, and a final extension step of 7 min/72 °C.

### 2.5. Sequence Determination of Infectious Disease Agents

The products obtained from all molecular assays were purified with a commercial kit (PureLink Quick Gel Extraction and PCR Purification Combo Kit; Invitrogen Life Technologies, Carlsbad, CA, USA), quantified using a Qubit Fluorometer (Invitrogen Life Technologies, Eugene, OR, USA), and then submitted for direct sequencing in both directions with the forward and reverse primers used in the respective molecular assays. Sequencing was performed using an ABI3500 Genetic Analyzer sequencer with the BigDye Terminator v3.1 Cycle Sequencing Kit (Applied Biosystems^®^, Foster City, CA, USA).

Sequence quality analyses and consensus sequences were obtained using the PHRED and CAP3 webpage (http://asparagin.cenargen.embrapa.br/phph, accessed on 8 January 2024), respectively. The identities of the nucleotide (nt) sequences obtained were compared with the nt sequences deposited in GenBank via BLAST (https://blast.ncbi.nlm.nih.gov/Blast.cgi, accessed on 8 January 2024) analysis. Selected sequences were deposited in GenBank. 

### 2.6. Determination of the Presence or Absence of Infectious Disease Agents

A farm or animal was considered to be infected when one or more of the agents investigated were detected at the farm or from the NS of the calf. Additionally, infections were considered singular when only one infectious disease agent was detected via a particular molecular investigation at each farm/animal. Similarly, infections were considered to be mixed/simultaneous when more than one of these agents was identified concomitantly at the farms and/or from the NS of the animal investigated.

### 2.7. Statistical Analyses

Microsoft Excel was used to store the data obtained and produce the graphical representations. The Epiinfo program (version 7.2.3.1) generated all frequency tables and was used to determine a possible statistically significant association between the molecular detection of disease pathogens of BRD and the occurrence of clinical manifestations. The Yates-corrected Chi-square or Fisher´s exact test was used with a significance level of *p* < 0.05. In addition, when appropriate, the obtained results were interpreted using descriptive statistical analyses.

## 3. Results

### 3.1. Animals, Sheep Rearing, and Infections

The calves from this study were derived from dairy cattle herds maintained in the rural areas of five cities; most of these are located within the East-Central mesoregion of Paraná state, except Clevelândia, located in the South-Central mesoregion (Table 1). All farms are enclosed dairy units with sheep rearing being performed within 0.5–1 km of Farms # 1 and 2, located on the outskirts of the city of Arapoti. However, the rural area of Castro has one of the largest sheep populations within the Paraná State [38]. Infections at these farms were associated with singular and concomitant associations due to *M. bovirhinis*, *P. multocida*, *M. haemolytica*, *H. somni*, BCoV, and OvGHV2 (see details below). Furthermore, the nucleic acids of *M. bovis*, other mollicutes, BVDV, BoAHV1, BRSV, and BPIV3 were not detected in the NS from any of the calves investigated during this study. A list of the agents identified in all calves during this study can be consulted (Appendix A Appendix A).

### 3.2. Molecular Amplification of M. bovirhinis and Other Infectious Agents

The *M. bovirhinis* nPCR assay was efficient for the specific nucleic acid amplification of this microorganism using the primers described in this study and resulted in the amplification of the desired bp of the ITS region. Similarly, the PCR and/or RT-PCR amplified the respective (amplicon) bp of the specific genes of *P. multocida*, *M. haemolytica*, *H. somni*, BCoV, and OvGHV2. 

Direct sequencing confirmed the nPCR assays of *M. bovirhinis* as well as the molecular assays of the other agents identified. Representative nt sequences of these agents are deposited in GenBank (*M. bovirhinis*, PP060739, PP060740, PP060741; OvGHV2, PP059653, and PP059654). 

### 3.3. Association between the Occurrence of Respiratory Infections in Asymptomatic and Diseased Dairy Calves

Only 6 of the 12 pathogens investigated were identified in calves during this study, resulting in infections due to *M. bovirhinis*, *P. multocida*, *M. haemolytica*, *H. somni*, BCoV, and OvGHV2. However, when the occurrence of respiratory infections was related to diseased and asymptomatic calves, a significant (*p* = 0.047) statistical association was only identified in calves reared at Farm #7 from the rural region of Castro that were infected by OvGHV2 (Table 1). Although *M. bovirhinis* was the most frequently identified pathogen in calves at all farms, resulting in infections in 57.8% (59/102) of the calves with clinical manifestations of BRD, as well as in asymptomatic calves (55.3%; 57/103), no significant statistical association (*p* = 0.718) was identified when these categories of calves were evaluated (Table 2). Other frequently occurring BRD pathogens diagnosed in dairy calves at these farms were BCoV, OvGHV2, and *Pasteurella multocida*, being identified in 93.3% (14/15), 80% (12/15), and 73.3% (11/15) of all farms investigated, respectively (Table 2). Of these three agents, BCoV was frequently diagnosed, occurring in diseased (52%; 53/102) and asymptomatic (51.5%; 53/103) calves. Similarly, infectious due to OvGHV2 occurred in diseased (37.2%; 38/102) as well as asymptomatic (27.2%;/28/103) calves. Nevertheless, there was no statistical association for the occurrence of these pathogens when the two categories of calves were compared.

### 3.4. Occurrence of Respiratory Pathogens in Dairy Calves with Clinical Manifestations of BRD

Calves with clinical manifestations of BRD had singular (*n* = 25), dual (*n* = 30), triple (*n* = 23), quadruple (*n* = 11), and quintuple (*n* = 4) infections due to the six agents diagnosed at these farms (Table 3 and Table 4). Singular infections were more frequently diagnosed in sick calves due to *M. bovirhinis* (32%; 8/25) and BCoV (28%; 7/25), followed by *P. multocida* (16%; 4/25) and OvGHV2 (12%; 3/25). Frequent dual infections (Table 4) in diseased calves were due to the simultaneous occurrences of *M. bovirhinis* + BCoV (36.7%; 11/30) and *M. bovirhinis* + OvGHV2 (30%; 9/30). Triple infections were more frequent due to the associations of *M. bovirhinis* (65.2%; 15/23) and OvGHV2 (52.2%; 12/23) with BCoV, *P. multocida*, *M. haemolytica*, and *H. somni*. Alternatively, quadruple infections occurred due to the frequent associations of OvGHV2 (81.8%; 9/11) and *M. bovirhinis* (63.6%; 7/11) with the other pathogens diagnosed (Table 4).

### 3.5. Frequency of Respiratory Pathogens in Asymptomatic Dairy Calves

As in diseased calves, their asymptomatic counterparts also had singular (*n* = 30), dual (*n* = 20), triple (*n* = 18), quadruple (*n* = 14), and quintuple (*n* = 5) infections (Table 3 and Table 4). However, infections due to *H. somni* were not identified in asymptomatic calves (Table 4). Frequent singular infections in asymptomatic calves were due to *M. bovirhinis* (56.7%; 17/30) and BCoV (23.3%; 7/30). The most frequent association of dual infections occurred in asymptomatic calves simultaneously infected with *M. bovirhinis* and BCoV (40%; 8/20). Triple infections were more frequent due to associations of BCoV (83.3%; 15/18), *M. bovirhinis* (72.2%; 13/18), *P. multocida* (66.7%; 12/18), and *M. haemolytica* (44.4%; 8/18). Curiously, quintuple infections were associated with the same combination of infectious agents (*M. bovirhinis* + OvGHV2 + BCoV + *P. multocida* + *M. haemolytica*) in asymptomatic (*n* = 5) and diseased (*n* = 4) calves (Table 4). 

### 3.6. Infection Dynamics Identified at Farms and in Diseased and Asymptomatic Calves from Southern Brazil

Calves at most farms had comparatively more mixed infections (*n* = 125) compared to singular (*n* = 55) infections via the six agents identified (Figure 1). Moreover, 12.2% (25/205) of all calves were not infected by any of the 12 pathogens investigated; 8.8% (9/102) of these being dairy calves with clinical manifestations of BRD, and 15.5% (16/103) were asymptomatic calves. When the agents associated with the singular and concomitant infections diagnosed in calves at these farms were analyzed, concomitant infections were the pattern identified only in calves from Farms #11 and 12. All other farms had a composition of singular and mixed infections, with mixed infections being predominant at Farm #2 (Figure 1). When all mixed infections were considered, BCoV (73.6%; 92/125) was the most frequently identified microorganism in association with another pathogen in the development of concomitant infections. Other agents with elevated frequencies of participation in simultaneous infections were *M. bovirhinis* (72.8%; 91/125) and OvGHV2 (48%; 60/125).

Furthermore, singular infections at these farms (Figure 2) were more frequent due to *M. bovirhinis* (45.4%; 25/55), followed by BCoV (25.4%; 14/55), *P. multocida* (10.9%; 6/55), and OvGHV2 (10.9%; 6/55). Additionally, singular infections via these four agents occurred only in calves at Farm #3, while singular infections via *M. bovirhinis*, BCoV, and OvGHV2 were identified in calves reared at Farms # 3 and 5. Infections associated with *H. somni* were only diagnosed in calves from Farm #4 (Figure 2). Calves from six farms (#1, 3, 5, 7, 8, and 13) were not infected by any of the 12 infectious agents evaluated, with the largest number of undiagnosed calves identified at Farm #3 (Figure 2). 

The interaction of the most frequently diagnosed pathogens (*M. bovirhinis*, BCoV, and OvGHV2) diagnosed in all calves is shown in Figure 3. These three pathogens occurred simultaneously in 10.7% (22/205) of all calves investigated during this study, with singular infections being more frequent to *M. bovirhinis* (16.6%; 34/205), followed by BCoV (13.7%; 28/205), and OvGHV2 (4.9%; 10/205). However, neither of these three pathogens was detected in the NS from 17.6% (36/205) of all calves investigated (Figure 3).

The interaction of the three most frequently diagnosed bacterial pathogens (*M. bovirhinis*, *P. multocida*, and *M. haemolytica*) in all calves is shown in Figure 4. *M. bovirhinis* was the most commonly identified in these animals, being diagnosed in 32.7% (67/205), with these three agents occurring concomitantly in 13.7% (28/205) of all calves investigated. However, 24.9% (51/205) of all calves were not infected by either of these three pathogens. 

## 4. Discussion

During this study, *M. bovirhinis* was the most frequently identified pathogen in all dairy herds investigated and from the NS of asymptomatic calves and calves with clinical manifestations of BRD. These findings are similar to the infectious trends associated with *M. bovirhinis* identified in North America [18,19,20] and Europe [21]. Other frequently identified pathogens were BCoV and OvGHV2, while the nucleic acids of several well-known agents (*M. bovis*, other mollicutes, BVDV, BoAHV1, BRSV, and BPIV3) of BRD investigated were not detected in any of the NS evaluated, suggesting that these pathogens were not associated with the clinical manifestations observed in the symptomatic calves from these farms at the time of sampling. Additionally, *P. multocida* and *M. haemolytica*, common bacterial pathogens associated with BRD [1], were not the predominant agents identified during this study. Similar findings were identified in dairy calves from Southern Brazil [39] and Denmark [11]. Alternatively, *P. multocida* was predominant within dairy calves from Southern Brazil [24], Finland [21], and Poland [40]. Furthermore, neither of these pathogens and only *M. bovis* and mollicutes were detected in a study from Southeastern Brazil [41]. The basic difference in these studies was the methodology used, with molecular diagnostics being performed in the investigations in Southern Brazil [39] and Europe [11,21,40], as opposed to the conventional bacteriological techniques (isolation and culture) performed in the study from Southeastern Brazil [41]. Consequently, these differences in the detection levels are primarily due to the elevated sensitivity and specificity associated with the molecular techniques compared to the traditional methods of bacterial culture and isolation. 

Furthermore, *H. somni*, the emerging multisystemic pathogen of ruminants in Brazil [42], was only identified in a relatively insignificant number of symptomatic calves maintained at a single farm. These findings align with a previous study performed in the same geographical location [24] and Finland [21]. Alternatively, the frequency of *H. somni* was elevated in dairy calves from another region of Southern Brazil [39], as was observed in European studies [11,40]. 

The results from this study suggest that most suckling calves (87.8%; 180/205) were either subclinically infected or developed clinical manifestations of BRD associated primarily with *M. bovirhinis*, BCoV, and OvGHV2, considering that the detection of these infectious agents in the NS of dairy calves without any associated demonstration of pulmonary disease confirms infection [43]. Of the three most frequent pathogens identified during this study, only BCoV is considered an inductor of BRD, while the participation of *M. bovirhinis* and OvGHV2 in pulmonary disease of cattle is not well established. Therefore, the ensuing discussion will be based on the possibility of these agents acting as respiratory pathogens of cattle.

The primers developed specifically for the diagnosis of *M. bovirhinis* proved to be efficient in the identification of this organism from the NS of calves with or without clinical manifestations of BRD. Therefore, these primers can be used to detect *M. bovirhinis* in cattle with respiratory disease. 

### 4.1. The Contribution of Myplasma bovirhinis in the Development of BRD

As far as the authors are knowledgeable, the results from this study represent the first investigation from Latin America to evaluate the possible participation of *M. bovirhinis* in the development of BRD. Our results demonstrated that there was no significant statistical difference between asymptomatic dairy calves relative to their counterparts with clinical manifestations of pulmonary impairment, suggesting that this microorganism may not be associated with BRD but a commensal of the respiratory tract of calves [8,10]. A closer evaluation of our results revealed that comparatively more asymptomatic calves (56.7%; 17/30) relative to calves with BRD (32%; 8/25) suffered from single infections of *M. bovirhinis* during this study; similar results were identified when the microbiota of calves with BRD was compared with asymptomatic calves [12], as well as in the healthy and pneumonic lungs of calves from the UK [10] and Finland [21]. These findings support the theory that *M. bovirhinis* should not be considered a “true” pathogen of BRD since the predominance of this microorganism within the nasopharyngeal microbiota may be attributed to its capacity to proliferate and evolve after weaned calves are maintained in feedlots [18]. Additionally, *M. bovirhinis* may simply be an opportunist invader of the respiratory system of calves with minimal role in the development of pulmonary diseases [44].

Alternatively, *M. bovirhinis* was associated with pneumonia in calves from Japan [13] and was the predominant organism identified using *q*PCR in an outbreak of BRD in beef calves from the USA [17]. The study performed in Japan isolated *M. bovirhinis* from the lungs of calves with suppurative pneumonia and detected antigens of *M. bovirhinis* by the production of hyperimmune serum [13]. However, the possible participation of other infectious agents associated with bacterial bronchopneumonia was not investigated, except *Pasteurella* spp. Furthermore, the study from Japan did not investigate the possible occurrence of simultaneous viral infections during the outbreak of respiratory disease, so the possible participation of these cannot be completely excluded. It must be highlighted that the utilization of PCR for diagnosing respiratory disease was not that popular when the Japanese study was published in 1973; therefore, molecular identification could have revealed additional agents of BRD. Alternatively, the study from the USA reported that there was no statistical difference between the bacterial load of *M. bovirhinis* identified within the lower respiratory tract when compared to calves with elevated body temperature, and all animals were simultaneously infected by BCoV [17]. Therefore, one wonders about the exact contribution of *M. bovirhinis* in the development of BRD in the study from the USA [17] since during our investigation, concomitant infections with *M. bovirhinis* and BCoV were responsible for (36.7%; 11/30) of all dual infections in calves with BRD. These two microorganisms also contributed to 40% (8/20) of the dual infections in the asymptomatic calves herein identified. Furthermore, *M. bovirhinis* was the second most frequently diagnosed microorganism (72.8%; 91/125) in all simultaneous infections identified during this study. 

Therefore, we propose that the effective participation of *M. bovirhinis* in the possible development of BRD should only be considered in singular infections. Collectively, there is not adequate information in the currently available published literature that effectively demonstrates the capacity of *M. bovirhinis* to serve as a primary pathogen of BRD. Additionally, the trend of *M. bovirhinis*-related infections identified in suckling calves from Southern Brazil during this study is similar to that observed in calves worldwide [12,14,21]. 

### 4.2. The Possible Participation of Ovine Gammaherpesvirus 2 in the Development of BRD

OvGHV2 is a *Macavirus* that causes sheep-associated-malignant catarrhal fever (SA-MCF) in a wide range of dead-end mammalian hosts worldwide [45,46]. Sheep is the asymptomatic host for OvGHV2, with infections in susceptible mammalian populations occurring predominantly due to contact with the nasal secretions of young lambs [46]. All *Macaviruses* known to cause malignant catarrhal fever (MCF) are collectively known as the MCF virus (MCFV) for sharing the 15A epitope [47]. Epidemiological data have demonstrated that most cattle with clinical SA-MCF develop pulmonary disease [45]. Furthermore, the initial replication of OvGH2 results in patchy and lytic interstitial pneumonia of experimentally infected sheep [48]. These factors were determinant in proposing an association between OvGHV2 and the development of BRD [45]. 

During this study, OvGHV2 was associated with the development of singular infections from asymptomatic (10%; 3/30) and diseased (12%; 3/25) calves, resulting in subclinical and clinical pulmonary infections, respectively, in the affected animals. These findings are in accord with the current pathogenesis of SA-MCF, where subclinical [49,50] and clinical [50,51] infections have been described in ruminants infected with OvGHV2. Additionally, OvGHV2 was indicated as the main cause of a pulmonary disease syndrome in an outbreak of dairy calves with clinical manifestations of BRD [7], while intralesional tissue antigens of a MCFV, more likely OvGHV2, were identified in the lungs of cattle with BRD [52], and pulmonary disease [53]. Furthermore, OvGHV2 participated in 48% (60/125) of all concomitant infections during this study. These findings are similar to the data obtained from a study that evaluated the detection of MCFV tissue antigens in cattle with BRD, where the MCFV, most likely OvGHV2, was identified in the lungs (53.3%; 64/120) of cattle that were infected via one or more agents [52].

Accordingly, it can be argued that the diseased and asymptomatic calves that were infected only by OvGHV2 during this investigation developed clinical and subclinical manifestations associated with this *Macavirus*. It must be highlighted that significant statistical differences were only identified when OvGHV2-associated infections were compared between asymptomatic and diseased calves at Farm #7, suggesting a direct association between infections by OvGHV2 and the development of BRD. Although there were statistical differences at this farm, the small number of animals sampled thereat warrants caution in interpreting these results. Additionally, sheep were only reared within 0.5–1 km of Farms #1 and 2, while infections associated with OvGHV2 were predominant during this study, occurring in 86.7% (13/15) of all herds, except for calves reared at Farms #13 and 15. Additionally, interstitial pneumonia was identified in outbreaks of SA-MCF in cattle from Northeatern Brazil [54]. These findings demonstrate that infections due to OvGHV2 are widespread in Brazil and are in accord with the results of a retrospective study that identified the elevated detection of antigens of MCFV in cattle with renal disease [55]. 

Consequently, the results from this investigation, in association with previous studies, support the theory that OvGHV2 should be considered an agent of respiratory disease in cattle. 

### 4.3. The Role of Bovine Coronavirus in the Development of Bovine Respiratory Disease 

The exact contribution of BCoV towards the development of classical singular respiratory infections remains controversial [56], primarily due to frustrating attempts to reproduce clinical disease [57]. Nevertheless, BCoV is frequently associated with respiratory infections in asymptomatic [24,58] and diseased [24,58,59,60,61] calves worldwide. However, there is adequate evidence to support the active participation of BCoV in the development of mixed respiratory infections in cattle [56,57,62]. These findings are in accord with the results of this study where BCoV was the second most frequently diagnosed microorganism associated with simultaneous infections. Furthermore, similar elevated frequencies of concomitant infections were identified in the NS [24] and the bronchial alveolar fluids [39] of asymptomatic and diseased dairy calves from Brazil. Moreover, BCoV was the main contributor towards the development of mixed infections in pneumonic lungs of beef cattle from Southern Brazil with elevated risk for BRD [61] but was not identified from a limited number of pulmonary samples derived from cattle maintained in a feedlot from Southeastern Brazil [63]. Collectively, BCoV is an endemic agent that is frequently associated with BRD in Brazil [2].

The elevated occurrence of respiratory field infections associated with BCoV worldwide is exacerbated due to concomitant infections and stress-related conditions [64]. It must be highlighted that the role of BCoV in the development of singular and concomitant respiratory infections in cattle has been extensively reviewed [56,57,62], and the overall consensus is that this agent is an established inductor of enteric disease in cattle. In contrast, the participation as a singular infectious agent associated with the development of BRD remains controversial. Alternatively, the simultaneous associations of BCoV with other respiratory pathogens seem more clinically significant in cattle with BRD [56,57]. Accordingly, the clinical impact of BCoV associated with BRD is more severe in a concomitant relative to singular infections in susceptible cattle populations. 

### 4.4. Study Limitations

Although the results are of significant importance to understanding the respiratory infection dynamics of *M. bovirhinis* in calves from Latin America, there were two limitations during the realization of this study that could have had some effect on the results presented herein: the repetition of samples and the type of sample. Only a single sample was collected from all calves, which may make identification of the exact stage of an ongoing infectious disease difficult. However, the determination of the phase of infection was not the focus of this study since the objective was to identify calves that were infected or not by the group of pathogens investigated. Although the utilization of NS for the detection agents of BRD is considered of reduced diagnostic value [65], there seems to be no significant difference in the detection of bacterial agents [66,67] and, to some extent, viral pathogens [67], of BRD when the different types of sample collection were evaluated. Additionally, the use of NS to diagnose BRD is recommended for identifying bacterial pathogens in diseased animals [68]. 

## 5. Conclusions

*Mycoplasma bovirhinis* was the most frequently diagnosed pathogen from the nasal swabs of asymptomatic and diseased dairy calves and contributed significantly towards the occurrence of concomitant infections during this investigation. However, the current accumulated data worldwide do not suggest that *M. bovirhinis* is a primary agent of BRD. At the same time, the infectious trends of *M. bovirhinis*-related respiratory infections diagnosed in calves from Brazil are similar to those described in other geographical locations. Furthermore, there is accumulating evidence to associate OvGHV2 with the occurrence of BRD, while the participation of BCoV as a singular agent of BRD remains controversial. 

## Figures and Tables

**Figure 1 pathogens-13-00114-f001:**
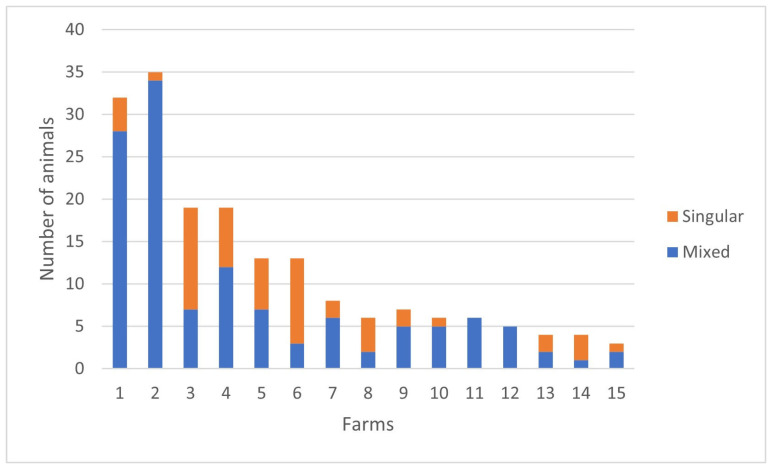
Frequency of singular and mixed infections diagnosed in dairy calves from Southern Brazil.

**Figure 2 pathogens-13-00114-f002:**
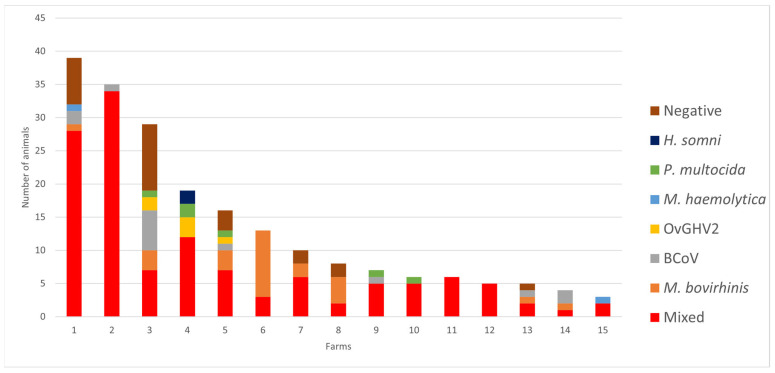
Distribution of singular and mixed infections in suckling calves and undiagnosed calves from dairy farms in Southern Brazil.

**Figure 3 pathogens-13-00114-f003:**
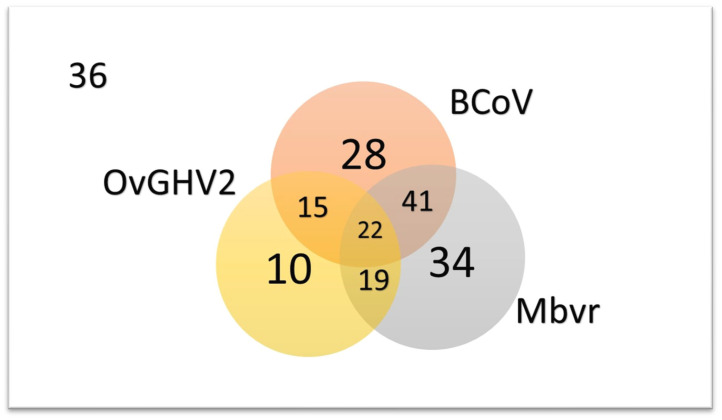
Occurrence of *Mycoplasma bovirhinis* (*Mbvr*), bovine coronavirus (BCoV), and ovine gammaherpesvirus 2 (OvGHV2) in asymptomatic and diseased suckling dairy calves from Southern Brazil. Legend: 36, number of calves not infected by these organisms.

**Figure 4 pathogens-13-00114-f004:**
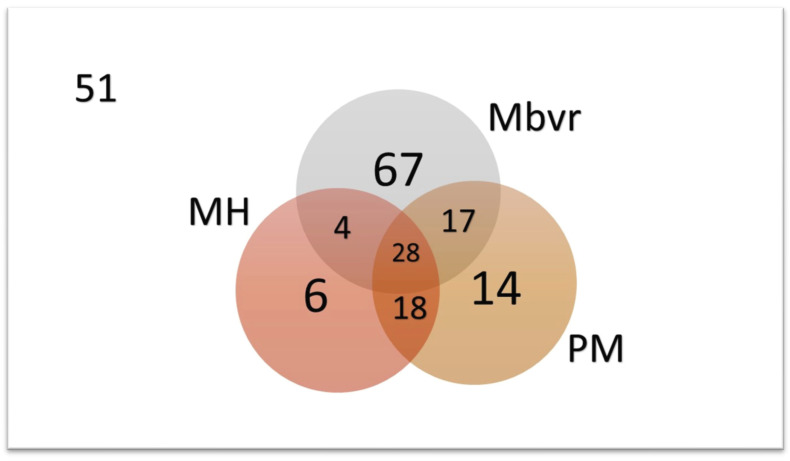
The interaction of *Mycoplasma bovirhinis* (*Mbvr*), *Pasteurella multocida* (PM), and *Mannheimia haemolytica* (MH) in asymptomatic and diseased suckling dairy calves from Southern Brazil. Legend: 51, number of calves not infected by these organisms.

**Table 1 pathogens-13-00114-t001:** Frequency of *Mycoplasma bovirhinis*, bovine coronavirus, and ovine gammaherpesvirus 2 diagnosed in nasal swabs derived from diseased and asymptomatic suckling dairy calves from the Paraná State, Southern Brazil ^1^.

Farms	Geographical Location		*Mycoplasma bovirhinis*			Bovine Coronavirus			Ovine Gammaherpesvirus 2		
	City	Mesoregion	+ve/dis (%)	+ve/asymp (%)	*p*-Value ^b^	+ve/dis (%)	+ve/asymp (%)	*p*-Value ^b^	+ve/dis (%)	+ve/asymp (%)	*p*-Value ^b^
1 (*n* = 39) ^a^	Arapoti	EC	6/13 (46.1)	11/26 (42.4)	1	9/13 (69.2)	17/26 (65.4)	1	4/13 (30.7)	9/26 (34.6)	1
2 (*n* = 35) ^a^	Arapoti	EC	10/19 (52.6)	9/16 (56.2)	1	17/19 (89.5)	15/16 (93.7)	1	10/19 (52.6)	3/16 (18.7)	0.086
3 (*n* = 29)	Arapoti	EC	3/12 (2)	7/17 (41.2)	0.612	5/12 (41.7)	6/17 (35.3)	1	1/12 (8.3)	4/17 (23.5)	0.370
4 (*n* = 19)	Clevelândia	SC	4/12 (33.3)	3/7 (42.7)	1	0/12 (0.0)	1/7 (14.3)	0.368	8/12 (66.7)	6/7 (85.7)	0.602
5 (*n* = 16)	Arapoti	EC	4/8 (50)	5/8 (62.5)	1	3/8 (37.5)	5/8 (62.5)	0.619	2/8 (25)	2/8 (50)	1
6 (*n* = 13)	Castro	EC	7/7 (100)	6/6 (100)	NPC	1/7 (14.3)	0/6 (0.0)	1	1/7 (14.3)	0/6 (0)	1
7 (*n* = 10)	Castro	EC	5/5 (100)	3/5 (60)	0.429	0/5 (0)	1/5 (20)	1	4/5 (80)	0/5 (0)	0.047
8 (*n* = 8)	Castro	EC	0/0	6/8 (75)	NPC	0/0	1/8 (12.5)	NPC	0/0	1/8 (12.5)	NPC
9 (*n* = 7)	Carambeí	EC	5/7 (71.4)	0/0	NPC	4/7 (57.1)	0/0	NPC	4/7 (57.1)	0/0	NPC
10 (*n* = 6)	Arapoti	EC	2/4 (50)	2/2 (100)	0.759	3/4 (66.7)	2/2 (100)	1	2/4 (50)	0/2 (0)	0.466
11 (*n* = 6)	Arapoti	EC	3/3 (100)	3/3 (100)	NPC	3/3 (100)	3/3 (100)	NPC	2/3 (66.7)	2/3 (66.7 )	1
12 (*n* = 5)	Carambeí	EC	5/5 (100)	0/0	NPC	5/5 (100)	0/0	NPC	0/5 (0)	0/0	NPC
13 (*n* = 5)	Carambeí	EC	2/2 (100)	1/3 (33.3)	0.576	2/2 (100)	1/3 (33.3)	0.400	0/2 (0)	0/3 (0)	NPC
14 (*n* = 4)	Carambeí	EC	1/2 (50)	1/2 (50)	1	1/2 (50)	1/2 (50)	1	0/2 (0)	1/2 (50)	1
15 (*n* = 3)	Castro	EC	2/3 (66.7)	0/0	NPC	0/3 (0.0)	0/0	NPC	0/3 (0)	0/0	NPC
Total			59/102 (57.8%)	57/103 (55.3%)	0.825	53/102 (51.9%)	53/103 (51.5%)	1	38/102 (37.2%)	28/103 (27.2%)	0.163

Legend: ^1^, data are only presented for the three most frequently diagnosed pathogens at these farms; ^a^, sheep were reared within 0.5–1 km from these farms; ^b^, *p*-values were obtained using the Yates-corrected Chi-square test or Fisher’s exact test with a significance level of *p* < 0.05; EC, Eastern Central; SC, Southern Central. NPC, not possible to calculate; +ve, positive; dis, diseased; asymp, asymptomatic.

**Table 2 pathogens-13-00114-t002:** Association of infectious agents diagnosed in diseased and asymptomatic suckling calves in dairy cattle herds from Southern Brazil.

Infectious Agents Identified	Number of Farms	Diseased Calves	Asymptomatic Calves	*p*-Value ^a^
*Mycoplasma bovirhinis*	15	57.8% (59/102)	55.3% (57/103)	0.718
Bovine coronavirus	14	52% (53/102)	51.4% (53/103)	0.949
Ovine gammaherpesvirus 2	12	37.2% (38/102)	27.2% (28/103)	0.124
*Pasteurella multocida*	11	41.2% (42/102)	34% (35/103)	0.289
*Mannheimia haemolytica*	3	24.5% (25/102)	30.1% (31/103)	0.371
*Histophilus somni*	1	5.9% (6/102)	0.0% (0/103)	0.965

Legend: ^a^ *p*-values were obtained using the Yates-corrected Chi-square test or Fisher’s exact test with a significance level of *p* < 0.05.

**Table 3 pathogens-13-00114-t003:** Occurrence of singular infections diagnosed in symptomatic and asymptomatic suckling dairy calves.

Infectious Disease Agent	Number of Calves	
	Symptomatic	Asymptomatic
*M* *ycoplasma* *bovirhinis*	8	17
Ovine gammaherpesvirus 2	3	3
Bovine coronavirus	7	7
*Pasteurella multocida*	4	2
*Mannheimia haemolytica*	1	1
*Histophilus somni*	2	0
Total	25	30

**Table 4 pathogens-13-00114-t004:** Occurrence of concomitant infections diagnosed in symptomatic and asymptomatic suckling dairy calves.

Infectious Disease Agents	Number of Calves	
	Symptomatic	Asymptomatic
Dual infections		
*M. bovirhinis*; OvGHV2	9	4
*M. bovirhinis*; BCoV	11	8
*M. bovirhinis*; *P. multocida*	1	1
*M. bovirhinis*; *M. haemolytica*	2	0
*M. bovirhinis*; *H. somni*	0	0
OvGHV2; BCoV	2	2
OvGHV2; *P. multocida*	0	2
OvGHV2; *M. haemolytica*	0	0
OvGHV2; *H. somni*	0	0
BCoV; *P. multocida*	1	0
BCoV; *M haemolytica*	2	3
BCoV; *H. somni*	0	0
*P. multocida*; *M. haemolytica*	0	0
*P. multocida*; *H. somni*	1	0
*M. haemolytica*; *H. somni*	0	0
Total	30	30
Triple Infections		
*M. bovirhinis*; OvGHV2; BCoV	4	5
*M. bovirhinis*; OvGHV2; *P. multocida*	2	1
*M. bovirhinis*; OvGHV2; *H. somni*	2	0
*M. bovirhinis*; BCoV; *P. multocida*	4	4
*M. bovirhinis*; BCoV; *M. haemolytica*	0	1
*M. bovirhinis*; *P. multocida*; *M. haemolytica*	3	2
OvGHV2; BCoV; *P. multocida*	2	0
OvGHV2; *P. multocida*; *H. somni*	2	0
BCoV; *P. multocida*; *M. haemolytica*	4	5
Total	23	18
Quadruple Infections		
*M. bovirhinis*; OvGHV2; BCoV; *P. multocida*	4	0
*M. bovirhinis*; OvGHV2; *P. multocida*; *M. haemolytica*	1	1
*M. bovirhinis*; BCoV; *P. multocida*; *M. haemolytica*	2	8
OvGHV2; BCoV; *P. multocida*; *M. haemolytica*	4	5
Total	11	14
Quintuple Infections		
*M. bovirhinis*; OvGHV2; BCoV; *P. multocida*; *M. haemolytica*	4	5
Total	4	5

Legend: *M. bovirhinis*, *Mycoplasma bovirhinis*; *P. multocida*, *Pasteurella multocida*; *M. haemolytica*, *Mannheimia haemolytica*; OvGHV2, ovine gammaherpesvirus 2; BCoV, bovine coronavirus.

## Data Availability

Nucleotide sequences for *M. bovirhinis* and OvGHV2 data from this study are deposited in GenBank.

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
