# Peer review of "The Role of Mycoplasma bovirhinis in the Development of Singular and Concomitant Respiratory Infections in Dairy Calves from Southern Brazil"

_pathogens, 2024, doi:10.3390/pathogens13020114_

Round 1

Reviewer 1 Report

Comments and Suggestions for Authors

This research adds to a growing body of knowledge that refutes the role of M. bvr as a primary pathogen of cattle.  While the authors provided a very extensive list of references. I think some key references are missing. Specifically, Chen et al (2018) are very forceful in stating that M. bvr is a pathogen and Miles et al (2004) suggested it might play a secondary role in BRD.   This paper deserves mention in the discussion. Conversely, ter Laak et al (1992), Ayling et al (2004), Thomas et al 2002, Maunsell and Donovan (2009) and Wolfger et al 2015 have all stated that there is little to no evidence of M. bvr being a pathogen. I think these references should be added and briefly discussed because they support the findings and conclusions of your study. 

My other comments are minor in nature:

Lines 24-26:  It is confusing to report in the same sentence prevalence based on the individual animal (xx/103) and then on the farm-level (xx/15).  I think there is value in providing both, since I am curious as to what percentage of farms had M. bvr.

Line 91 - include your cut-off value in degrees for what constituted a fever - was it >40C?

Line 146 - the descriptive statistics could have been improved by having a table showing number of animals by quartiles that were infected. Gives a bit more information that simple herd-level percentages.

Line 227-229. Reporting of percentages is confusing because the denominator changes from 205, to 102 and 103. I think just stating that the 12 pathogens were not detected in 25 calves, 9 of which were clinical cases and 15 asymptomatic. 

The Discussion should not any limitations or weaknesses of the study. There are always limitations.

Table 1 - the p-value of 1 should be consistently reported to 3 decimal places, ie. 1.000.  This is also an issue in Table 2 were you report to 4 decimal places.  This needs to be consistent.

Table 1 - NPC.  It is not uncommon to add 0.5 to all the cells in your chi-square table to generate a p-value. This just needs to be explained in the methods.

Figures 1 and 2.  These figures do not add a lot and should be moved to the supplementary file. 

Comments on the Quality of English Language

Line 215 - rewrite --  Histophilus somni was not recovered from any asymptomatic calves. 

Reviewer 2 Report

Comments and Suggestions for Authors

The authors present a study where they have set out to determine if Mycoplasma bovirhinis is associated with bovine respiratory disease (BRD) in suckling dairy calves. Calves were assessed for clinical signs indicative of BRD and nasal swabs were collected. Nasal swabs were also collected from calves free of clinical signs. Following nucleic acid extraction, the swabs were tested for the presence of M. bovirhinis and bacterial, viral, and mycoplasma species using conventional PCR assays. The authors have subsequently analysed the data for associations between the detection of these agents and BRD.

Somewhat the only significant association detected in the study was for ovine gammaherpesvirus 2 (OGHV-2) on one farm. For the remainder of the farms tested no associations between the microbes of interest and BRD were detected. Rather, the numbers of calves positive for each agent were quite similar between those calves deemed to have BRD and those calves deemed to be healthy controls. While a significant association is reported for OGHV-2 in diseased calves from Farm 7, it is not clear how this association was determined. As the numbers in the cases and controls are five each, even with four positives in the BRD cases and none in the controls, I could not find a significant association.

The methods section also suggests that odds ratios were determined in the study. Although none are presented in the study.

In revising their manuscript, the authors should ensure that the methodologies used in their statistical analyses are clear. The approaches used could be added to the relevant results tables either in the titles or as footnotes.

One interesting feature of this study the even numbers of positives for the microbes of interest in the samples from BRD cases and controls, as shown in Table 1. These results further highlight the need for clear case definitions as discussed above. As an example, one of the issues with single point-in-time testing is that it can be difficult to determine what stage of disease development the animal being assessed is at. If the animal tests positive to the agent of interest and is also exhibiting clinical signs, then it is reasonable to make some assumptions about the role of the organism in the disease of interest. However, if an animal is free of clinical signs used to diagnose the disease and tests positive for the microorganism targeted by the diagnostic assay, it is plausible that the animal is in the early stages of disease development and if additional samples were collected or monitoring conducted, the control may end up being a case. This is particularly relevant for diseases such as BRD where a multitude of factors impact the risk of animal developing disease. These aspects should be discussed in the discussion.

Some parts of the current discussion are long and consideration should be given to shortening it.

Line 2 suggest replacing “participation” with “role”

Line 19 I would suggest the authors add one or two sentences at the start of the abstract to introduce the problem the study aims to address. It currently launches straight into the study and does not provide any context for it.

Line 19 Here and throughout the manuscript, I would suggest the authors adopt the conventional use of abbreviating species names for Mycoplasma bovirhinis as M. bovirhinis.

Line 25 I would suggest rounding these estimates to one decimal place here and throughout the manuscript. Two decimal places suggest a precision that would be difficult to justify.

Line 29 Suggest revising the text by adding in the number of affected calves, as “few” is subjective.

Line 88 Please add a brief description of the nasal swabs used in the study.

Lines 90-92 I would suggest the authors tighten their case definition here. They describe cases as being those calves exhibiting at least four of the five listed clinical signs. Control animals were defined as those calves that exhibited none of these signs. These definitions raise the question of how animals with one to three of these clinical signs were classified. Please review this text and revise as necessary to ensure animal classifications are clear.

Line 91 How was “fever” defined/determined?

Line 103 – Does the BVDV assay detect all species of the virus?

Line 145 The authors state in this text that an animal was considered infected if the bacterium, mollicute, or virus was detected by PCR. The use of the term “infected” is suggestive of disease. It is generally accepted that many of the bacteria and possibly the mollicutes are present in the upper respiratory tract of cattle as components of the normal microbiota in the absence of disease. Similarly, some studies have reported the detection of some viruses in healthy and diseased cattle. The point is that I am not sure it is valid to deem animals as infected through detecting a microbe by PCR. I think it would be more accurate to classify this outcome as positive/negative or present/absent.

Line 382 suggest replacing “incriminated” with “associated”

Line 421 Table 1

As mentioned previously the statistical methods used in determining the p values should be described.

Did the authors investigate if collapsing the results by “city” yielded any meaningful associations?

The authors should consider providing the results of all the PCR analyses as a supplemental file. Particularly results other than those shown in Table 1 are presented as cumulative results in subsequent tables and figures.

Line 425 Table 2

Suggest providing some details on what statistical analysis was done.

Table 3 and Table 4

I would suggest splitting these tables and then combining them with like for like. For example, the new Table 3 would have single infections for both diseased and healthy animals, Table 4 would have dual infections for both diseased and healthy animals, etc for the remaining combinations. I think this is more meaningful as it allows direct comparisons to be made between the healthy and diseased groups. After all, this is the key point of interest – what is different between healthy and diseased animals?

Round 2

Reviewer 2 Report

Comments and Suggestions for Authors

The authors have addressed the comments and answered the questions I raised in my review of the submitted version of their manuscript. 

I have no comments on the revised manuscript.